

# Variation in the reproductive quality of honey bee males affects their age of flight attempt

Bradley N. Metz and  David R. Tarpy

Applied Ecology, North Carolina State University, Raleigh, NC, United States of America

## ABSTRACT

**Background**. Honey bee males (drones) exhibit life histories that enable a high potential for pre- or post-copulatory sperm competition. With a numerical sex ratio of ~11,000 drones for every queen, they patrol flyways and congregate aerially to mate on the wing. However, colonies and in fact drones themselves may benefit from a relative lack of competition, as queens are highly polyandrous, and colonies have an adaptive advantage when headed by queens that are multiply mated. Previous research has shown that larger drones are more likely to be found at drone congregation areas, more likely to mate successfully, and obtain a higher paternity share. However, the reproductive quality and size of drones varies widely within and among colonies, suggesting adaptive maintenance of drone quality variation at different levels of selection.

**Methods**. We collected drones from six colony sources over the course of five days. We paint marked and individually tagged drones after taking body measurements at emergence and then placed the drones in one of two foster colonies. Using an entrance cage, we collected drones daily as they attempted flight. We collected 2,420 drones live or dead, analyzed 1,891 for attempted flight, collected emergence data on 207 drones, and dissected 565 upon capture to assess reproductive maturity. We measured drone body mass, head width, and thorax width at emergence, and upon dissection we further measured thorax mass, seminal vesicle length, mucus gland length, sperm count, and sperm viability from the seminal vesicles.

**Results**. We found that drones that were more massive at emergence were larger and more fecund upon capture, suggesting that they are of higher reproductive quality and therefore do not exhibit a trade-off between size and fecundity. However, smaller drones tended to attempt initial flight at a younger age, which suggests a size trade-off not with fecundity but rather developmental maturation. We conclude that smaller drones may take more mating flights, each individually with a lower chance of success but thereby increasing their overall fitness. In doing so, the temporal spread of mating attempts of a single generation of drones within a given colony increases colony-level chances of mating with nearby queens, suggesting an adaptive rationale for high variation among drone reproductive quality within colonies.

Corresponding author
Bradley N. Metz, bnmetz@ncsu.edu

# INTRODUCTION

Like their sister workers and queens, honey bee (*Apis mellifera* L.) drones have little control over their rearing conditions, with colony-level factors (such as worker population), resource availability, and season all impacting the likelihood for a colony to rear drones or reject them (*Boes, 2010*; *Seeley & Mikheyev, 2003*; *Smith et al., 2014*). For adult drones, fecundity measures (*e.g.*, sperm count and viability) and size measures (*e.g.*, wing length, body mass) are almost always positively correlated (reviewed in *Metz & Tarpy, 2019*). Further, body size is a major determinant of mating success, with larger drones tending to be more successful at mating, delivering more spermatozoa, and obtaining a higher proportion of worker paternity (*Couvillon et al., 2010*). It would therefore seem that larger drones would exhibit higher fitness, which would suggest directional selection for large drones and therefore low variation among males within a colony. This, then, raises the question: why do drones vary widely in their size and fecundity?

There are several factors that can explain the high variation in drones based on their parental origin and how they are reared. First, early examinations explored the potential for honey bee workers to lay male eggs (*e.g.*, *Gençer & Kahya, 2011*). However, worker-laid drones represent only a tiny fraction of those produced (*Visscher, 1989*), likely owing to policing of worker reproduction (*Pirk et al., 2004*). Second, mother queens may mistakenly lay drone eggs into worker cells. However, genetic analyses of queen-laid eggs showed that under normal circumstances, the rate of a queen laying unfertilized eggs into a worker cell is extremely low (*Ratnieks & Keller, 1998*). Third, there is variation in nurse attention to worker larvae that has potential consequences for their fates as adults, leaving them less likely to be selected for queen rearing (*Sagili et al., 2018*) or less effective foragers (*Scofield & Mattila, 2015*). Similarly, different drone larvae may express parasitic feeding or begging behavior (*e.g.*, *He et al., 2016*; *Peignier et al., 2019*), resulting in differential feeding and thus larger growth compared to their nestmates. To our knowledge, nursing variation towards drones has not been observed, but these proximal causes of drone variation cannot be ruled out and deserve additional study. Finally, colony nutritional status may not impact just the amount of brood or the number of drones reared (*Ahmad et al., 2021*; *Brodschneider & Crailsheim, 2010*; *Hoover, Ovinge & Kearns, 2022*; *Noordyke & Ellis, 2021*), but also influence their relative fecundity since food limitation may impact the provisioning of brood food to drone larvae.

Moreover, drone quality may vary as a function of being haplodiploid individuals more susceptible to colony stresses. Drone reproduction, survival, and maturation are negatively impacted by a host of factors, such as season, temperature stress, pollen deprivation and beekeeper-applied acaricides or farmer-applied insecticides (*reviewed in Rangel & Fisher, 2019*). Further, drones may be more susceptible to pesticide (*Fisher & Rangel, 2018*; *Friedli et al., 2020*; *Grassl et al., 2018*; *Kairo et al., 2016*; *McAfee et al., 2022*) or pathogen challenge (*Retschnig et al., 2014*) than workers. In familial groups with high inclusive fitness, there is potential for conflict between individual and colony-level selection such that traits promoting the fitness of the colony may come at a cost of those promoting individual fitness or vice versa (*Ratnieks & Reeve, 1992*; *Reeve & Keller, 1999*). Additionally, because

of their haploid genetic structure, drone phenotypes are likely to be constrained by genetic architecture contributing to the worker and queen phenotypes (*Rueppell, Page & Fondrk, 2006*), weakening the effect of selection on drones and resulting in a greater variation in size and fecundity than might be otherwise expected.

As opposed to being a consequence of social or parentage factors, or resulting from weakened directional selection, variation in drone quality may instead be maintained by adaptive selection. For instance, in parallel to the model of intra-ejaculate trade-offs of individuals (*Decanini, Wong & Dowling, 2013*), some colonies may favor the production of more, smaller drones, rather than fewer, larger ones. In this case, we would expect to see that the mean size of drones produced by a colony would covary with the number of drones that colony reared, something that has yet to be tested. Intracolonial variation of drones may also be adaptive if drones of various sizes perform different individual-level sexual strategies (*e.g.*, smaller males fly at younger age, earlier in the day, or closer to the colony), in this case a high degree of intracolony variation would be expected to be maintained overall. The influence of size on male sexual strategy is widespread among animals (*Shuster, 2010*) with males with less direct competitive ability taking advantage of alternate strategies (*e.g.*, *Gross, 1996*; *Nason & Kelly, 2020* and references therein).

In an example from solitary bees, Eastern carpenter bee *Xylocopa virginica* males either defend a territory or patrol areas around other males' territories as interlopers (*Barrows, 1983*) and the likelihood to perform either behavior is based at least partially on size (*Duff, 2018*). This is simlarly true for the wool-carder bee *Anthidium manicatum* (*Severinghaus, Kurtak & Eickwort, 1981*), and males of *Centris pallida* alternatively dig in the ground for female emergence sites when larger or patrol the vegetation for females when smaller (*Alcock, 2013*). Finally, in a stingless bee *Scaptotrigona* aff. *depilis*, smaller individuals remained in mating aggregations for longer periods of time (*Koffler et al. , 2016*). Honey bee drones are produced in extreme numbers, with an approximate 11,000:1 mating ratio found at drone congregation areas (*Koeniger et al., 2005*; *Page & Metcalf, 1984*). However, mating takes place without obvious aggression, with drones assembling in a ''comet'' behind the queen, with one after another darting forward to mate and drones preferring queens that have already been mated (*Gries & Koeniger, 1996*; *Koeniger, 1990*). Drones reared from worker cells are more likely to fly outside of the times of peak mating activity compared to normal-sized males (*Couvillon et al., 2010*) suggesting the possibility of different individual-level mating strategy (although this is contradicted by earlier work by *Berg et al., 1997*). However, smaller drones are less likely to be found at drone congregation areas (*Berg, 1991*), and there is conflict over whether drone congregation areas are truly sites of mating in a natural setting or are instead convenient locations for researchers to find drones (*Loper, Wolf & Taylor Jr, 1992*). This may point to drones of varying quality exhibiting differential mating strategies.

When randomly sampling from a flight-restricted population of drones, we found that older drones tended to be smaller, which would suggest that smaller drones live longer (*Metz & Tarpy, 2019*; but see *Czekońska, Szentgyörgyi & Tofilski, 2018* for a potentially contradictory result). Drones begin taking mating flights approximately 11 days after adult emergence, although drones that initiate flight at younger ages tend to make more

flights over the course of their lives (*Rueppell et al., 2005*). However, despite a wealth of information on variation in flight ontogeny in honey bee females (*Rueppell et al., 2004*) and evidence that drones utilize similar physiological mechanisms (*Giray & Robinson, 1996*), there is, to our knowledge, no study directly assessing variation in flight initiation of drones as it relates to body size or reproductive measures. Herein, we explore the variation in flight and reproductive ontogeny in drones with the hypothesis that small drones may initiate flight at earlier ages, therefore potentially making more flights, balancing the relatively lower chance of a successful mating attempt with the maximization of mating attempts.

## MATERIALS AND METHODS

### Drone rearing

A single standard-deep frame (243 mm × 480 mm) of emerging drones was collected from each of six colony sources headed by Italian queens at the Lake Wheeler Honey Bee Research Facility in Raleigh, North Carolina, USA. Queens were reared and open-mated in our uncontrolled local population, and colonies were maintained by standard beekeeping practice prior to experimentation. Drone eggs were laid by the queens into selected and measured frames of drone-sized comb. A ''mite count'' (standard sampling of *Varroa destructor*) was taken by powdered sugar shake using approximately 300 bees based on a 0.5 cup volume (*Macedo, Wu & Ellis, 2002*). Mean cell size of each drone frame was estimated by counting three rows of 20 cells and averaging the width. Frames were removed from their colonies two days before expected emergence and stored in individual boxes in an incubator at 33 °C and ∼50% Relative Humidity along with 100 workers from the brood nest of the source colony to aid in emergence.

Each day at 6:00 and 18:00, emerged drones were captured and marked with paint; each drone was marked on the abdomen with a unique color according to its source colony and on the thorax according to its day of emergence. Because prior research suggests that adult colony environment impacts age of first foraging in workers (*Winston & Katz, 1982*), we transferred all drones from source colonies to separate foster colonies. Drones emerging in the morning and evening were then taken to separate colonies for fostering to keep the cohort as closely age matched as possible for the estimation of age of first flight. Drones that emerged from 18:00-6:00 (and placed into one foster colony:AM) were considered to be 12 h (0.5 days) older than the drones that emerged from 6:00–18:00 (and placed into a separate foster colony:PM) on the same day. Marked drones were lightly shaken onto the top bars of the frames of the foster colonies to introduce them. Emergence and marking proceeded from 06/17/18-06/24/18 , since we endeavored to minimize the potential seasonal variation in drone quality. On 06/22/18 at 6:00 instead of paint-marking, 214 emerged drones were individually number tagged, weighed, and photographed prior to being added to their respective foster colony. We did this so that we could better measure individual drones for all phenotypes rather than estimating age cohorts only. Only five of the six source colonies had drones that emerged on this date.

Foster colonies were selected as being similar in size and condition to each other and the source colonies, and they were placed adjacent to each other within the same apiary.

Each colony was housed in a hive consisting of two standard Langstroth brood boxes with a top feeder regularly supplied with 1:1 sucrose:water solution. To estimate age of first flight, we collected drones as they first left the nest into a size-selecting trap. This method resulted in an estimation of flight that may include drones being expelled from the nest and drones taking sanitation or orientation flights prior to mating but ensured that we were highly unlikely to miss focal drones and bias the sample against those drones that successfully mate. Colonies were fitted with a custom-built drone trap consisting of cleaned plywood with queen excluder to prevent large drones from escaping. Drone traps had a 7.6 cm hole on the top and a wire mesh funnel with a 1.3 cm opening leading into a secondary cage made of plywood and queen excluder (Fig. 1). Drones that attempted to leave the nest flew up into the cage, were prevented from exiting, and were unable to return to the nest. Drones were collected from the cages daily at 18:00, after the bulk of mating flights for the day were concluded, and immediately transported to the lab for dissection and measurement. Occasionally, drones were found dead in the bottom of the cage; these were not always distinguishable as having flown or having been carried by undertakers, and they were counted separately but not dissected. Finally, counts of dead drones in the trap bottom were taken every other day in the morning prior to the period of drone flight activity but were excluded from any subsequent analyses.

## Dissection and measurement

The number of drones captured quickly exceeded our ability to fully analyze their reproductive quality. As such, all paint-marked drones were weighed and counted for age and colony source, but once the number of drones exceeded 30 per trap only one of each age- and colony pairing was dissected and measured as previously reported (*Metz & Tarpy, 2019*). All individually tagged drones, however, were dissected and fully analyzed. Briefly, each was weighed to the nearest 0.1 mg and its head and thorax photographed. They were then dissected with the mucus glands and seminal vesicles removed, cut free from the testicles and ejaculatory duct, which were also photographed. Finally, the head, wings, legs, and abdomen were cut free from the thorax, which were weighed. The seminal vesicles were immediately ruptured in 1.0 mL of a saline buffer, Buffer D (*Collins & Donoghue, 1999*; *Makarevich et al., 2010*; *Metz & Tarpy, 2019*) and lightly mixed to homogenize. This obviated the need for the drone to be capable of ejaculating to assess sperm viability. The solution was then dyed using the Invitrogen live/dead spermatozoa staining kit # L7011 (Carlsbad, CA) and read using a Nexcelom Cellometer$^®$ Vision Sperm Counter machine (Nexcelom Bioscience LLC; Lawrence, MA, USA) to gain a count of viable sperm. The photographs were then analyzed using ImageJ version 1.51m9 (*Schneider, Rasband & Eliceiri, 2012*) to measure the width of the head and thorax (as measured by the distance between tegulae), as well as the length of the seminal vesicles and mucus glands.

## Statistical analyses

Experimental methods resulted in multiple populations of drones, for which there was varying levels of information. At minimum, all analyzed drones were associated with colony source, colony mite count, mean drone cell size, foster colony, emergence date, age when

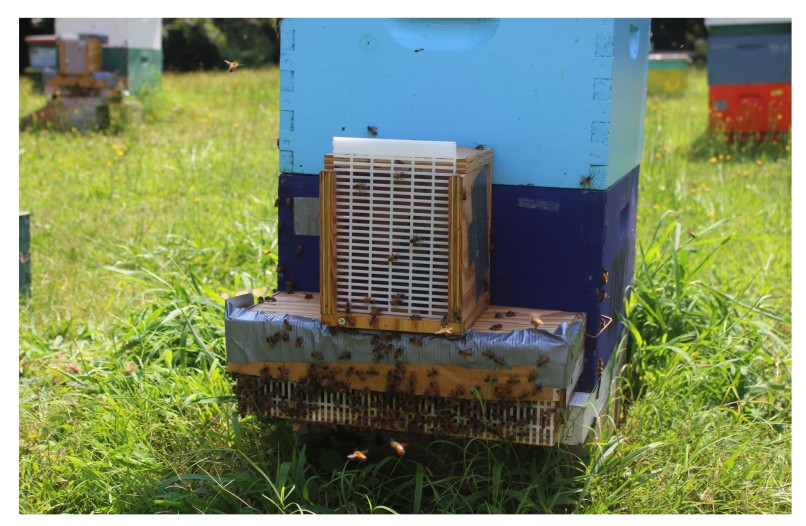

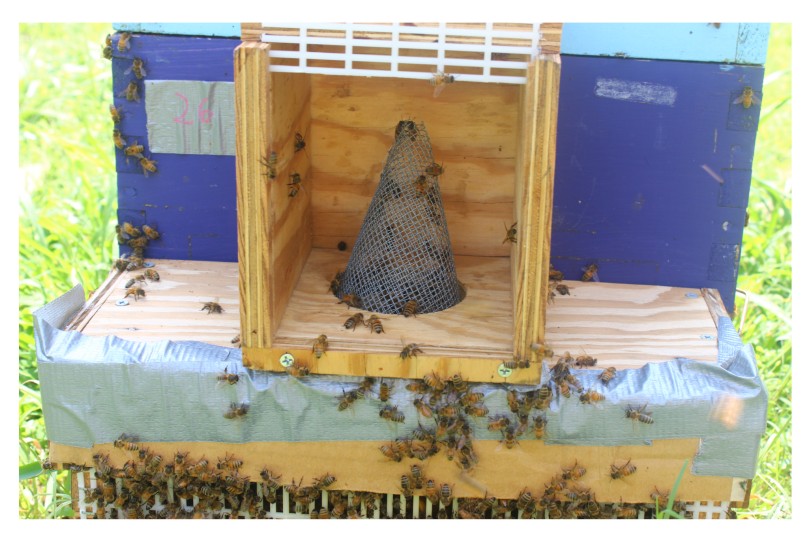

**Figure 1 Drone trap construction and placement.** The flight cages used for this study. The bottom part of the trap consisted of a runway and queen excluder material, which restricted drone movement, while the upper consisted of a mesh cone and large face of excluder. Drones, unable to exit through the bottom, were attracted into the top of the trap due to the light and were unable to return into the bottom. Photos taken by JP Milone.

trapped, location trapped, and condition (live/dead). All live drones at capture included their weight as well, and all drones were recorded for estimation of age of first flight based on their known emergence and collection dates. Those drones that were captured alive and dissected were also assessed for fecundity and size measures, as described above. Finally, the individually marked drones had all the above variables plus their emergence mass, thorax width, and head width associated with their age of first flight. Experimenters were not blinded to the colony identity or emergence date of the drones during data collection or analyses. Statistical analyses and visualization were performed using R (version 3.6.0;

**Table 1 Statistical packages for R used in these analyses.**

| Title | Use in these analyses | Citation |
|---|---|---|
| agricolae | Post-hoc analyses for linear model | *de Mendiburu (2021)* |
| broom | Data handling particularly for linear models | *Robinson, Hayes & Couch (2020)* |
| cowplot | Annotation of multi-part figures | *Wilke (2020)* |
| gridExtra | Arrangement of multi-part figures | *Auguie & Antonov (2017)* |
| gtsummary | Data handling of statistical results reporting | *Sjoberg et al. (2021)* |
| Hmisc | Spearman's correlation calculations | *Harrell Jr et al. (2021)* |
| readxl | Import and process raw data files | *Wickham & Bryan (2019)* |
| survival | Survival analyses calculation | *Therneau (2021)* |
| survminer | Visualization of survival analyses results and hazard ratios | *Kassambara, Kosinski & Biecek (2021)* |
| tidyverse | Data handling, descriptive statistics, and visualization | *Wickham (2019)* |

*R Core Team, 2019*) with relevant packages referenced in Table 1. Statistical code and the dataset used in these analyses are provided as Supplemental Information.

# RESULTS

## Collection summary

We collected attempted-flight data from a total of 1,891 drones. We individually tagged 207 drones and recovered 70 live for dissection and a recovered a further 77 dead in either the collection cage or trap bottom. The remainder were lost, likely due to the removal of the number tags. We dissected an additional 495 paint-marked drones and counted then weighed 887 live drones collected from the drone trap. Finally, we collected 831 dead drones from the cage or trap bottom. Distribution of collected drones and their use in these analyses are presented in Tables 2–4.

## Age of capture differs by colony source

This analysis included all drones collected in the study. We defined flight attempt as drones being found live in the upper cage of the trap, although it is possible (albeit unlikely) that dead drones were carried there by workers expelling them from the nest. Since trapped drones are not fed by workers and can quickly die in confinement, all drones (live or dead) found in the upper cage of the trap were considered to have attempted flight. Dead drones in the upper cage were censored such that they were assumed to have flown on the date collected or later. Drones collected (dead) in the lower part of the trap were excluded from these analyses. Tagged drones that were lost were similarly not included (Table 4).

Median age of capture for drones from each source are presented in Figs. 2A and 2B, and the hazard ratios (HR) are presented in Figs. 2C and 2D. The global median age at which drones were found in the traps was 8.5 days, ranging from 0.5 d to 37 d. An initial Cox proportional hazards model containing colony source (a combination of genetics and rearing environment) and foster colony (adult environment) showed that adding an interaction term among foster colony and source colony significantly improved model fit ($Chi^2_5 = 14.9$; $p = 0.011$). We therefore analyzed each foster colony separately. Source was

**Table 2  Distribution of the fates of individually tagged drones.** Live drones were captured live in the cage and dissected. Dead drones were found dead in either the cage or the trap bottom. Lost drones were tagged upon emergence, but not recovered.

| Source colony | Live | Dead | Lost |
|---|---|---|---|
| A | 6 | 20 | 6 |
| C | 7 | 0 | 5 |
| D | 36 | 44 | 26 |
| E | 9 | 7 | 18 |
| F | 12 | 6 | 5 |

**Table 3  Distribution of the fates of paint-marked drones.** Drones were collected and for each dissection day and foster colony, one of each source colony and age pairing was dissected to assess size and fecundity parameters. The remaining drones were weighed and tallied. Dead drones were found dead in either the cage or the trap bottom and counted.

| Source colony | Foster colony | Dissected | Tallied | Dead |
|---|---|---|---|---|
| A | AM | 41 | 169 | 340 |
|   | PM | 55 | 93 | 117 |
| B | AM | 24 | 42 | 39 |
|   | PM | 24 | 50 | 15 |
| C | AM | 26 | 15 | 23 |
|   | PM | 20 | 11 | 11 |
| D | AM | 84 | 188 | 109 |
|   | PM | 69 | 108 | 62 |
| E | AM | 46 | 84 | 64 |
|   | PM | 44 | 14 | 16 |
| F | AM | 21 | 78 | 14 |
|   | PM | 41 | 35 | 21 |

a significant factor in the variation in likelihood to fly. In the first foster colony, colonies D (HR = 0.62; $p < 0.001$) and E (HR = 0.58; $p < 0.001$) had significantly lower likelihood to fly (and therefore were found in the trap at a significantly older age) compared to colony A (Fig. 2C). In the second foster colony, colonies C (HR = 0.65; $p = 0.028$), D (HR = 0.48; $p < 0,.001$), E (HR = 0.50; $p < 0.001$), and F (HR = 0.63; $p < 0.001$) all had significantly lower risk to fly compared to colony A (Fig. 2D).

## Drone emergence characteristics and age of capture

We measured the subset of drones that were tagged and recaptured to observe effects of different drone size measures at emergence on age of capture and measures at capture (Table 2). Drones emerged with a body mass of $220.3 \pm 26.2$ mg, head width of $4.31 \pm 0.13$ mm, and thorax width of $5.46 \pm 0.24$ mm. We tested first for colony-level differences in emergence properties using univariate ANOVA. We found that emergence size differed among the colonies, with head width ($F_{4,200} = 35.0$; $p < 0.0001$), thorax width ($F_{4,200} = 14.0$; $p < 0.0001$), and mass ($F_{4,202} = 60.9$, $p < 0.0001$), all being significant (Figs. 3A–3C).

**Table 4 Distribution of drones analyzed for estimation of age of attempted flight.** Number of drones that were captured live in the cage top (either individually tagged or paint marked) were considered to have attempted flight. Censored drones found dead in the cage top were considered to have attempted flight at the age they were collected or older. Excluded drones were found dead in the trap bottom and not analyzed for flight attempt estimation.

| Source colony | Foster colony | Attempted flight | Censored | Excluded |
|---|---|---|---|---|
| A | AM | 216 | 118 | 248 |
|   | PM | 148 | 57 | 60 |
| B | AM | 66 | 24 | 15 |
|   | PM | 74 | 7 | 8 |
| C | AM | 48 | 13 | 15 |
|   | PM | 31 | 8 | 3 |
| D | AM | 308 | 96 | 83 |
|   | PM | 177 | 42 | 20 |
| E | AM | 139 | 41 | 48 |
|   | PM | 58 | 8 | 8 |
| F | AM | 111 | 11 | 14 |
|   | PM | 76 | 14 | 7 |

We then tested each emergence measure along with colony source in a Cox proportional hazards model, finding a significant effect of colony source with drones from colony C having a significantly higher risk to fly (HR = 9.5; $p < 0.001$; Fig. 3D), consistent with the results from the paint-marked bees, and emergent body mass such that more massive drones had a slightly lower risk to fly than their lighter brothers (HR = 0.97; $p = 0.004$; Fig. 3E).

## Causes and correlates of drone variation in emergence size and age of capture

To explore potential causes of colony-level variation in drones, the average cell size of drone frames was estimated, and a mite count of the source colony was taken when the frame was removed. Mean cell size of the drone frames ranged from 12.88–13.93 mm, and mite counts of drone-source colonies ranged from 3–21. We tested for a relationship between emergence measures and colony characteristics using linear regression. Mite count had a slight but significant effect on drone emergent mass, with generally less-massive drones coming from colonies with more mites ($F_{1,205} = 8.9$; $p = 0.003$; $r^2 = 0.037$) but not on head width ($F_{1,203} = 0.40$; $p = 0.52$) or thorax width ($F_{1,203} = 0.72$; $p = 0.40$). Cell size, on the other hand, had a slight but significant effect on all three measures. Unexpectedly, this was a negative relationship, with larger cells producing slightly less-massive drones ($F_{1,212} = 23.6$; $p < 0.0001$; $r^2 = 0.096$) with narrower thorax ($F_{1,210} = 24.9$; $p < 0.0001$; $r^2 = 0.10$) and head widths ($F_{1,210} = 38.26$; $p < 0.0001$; $r^2 = 0.15$). Mite count and cell size were not significantly correlated (Spearman's Rho = −0.029; $p = 1$).

Expecting that drones would lose mass from emergence to capture, we then calculated the difference in body mass, head width, and thorax width and used a *t*-test to determine significance from zero. We found that drones lost an average of 21.51 ± 2.28 (9.4%) mg

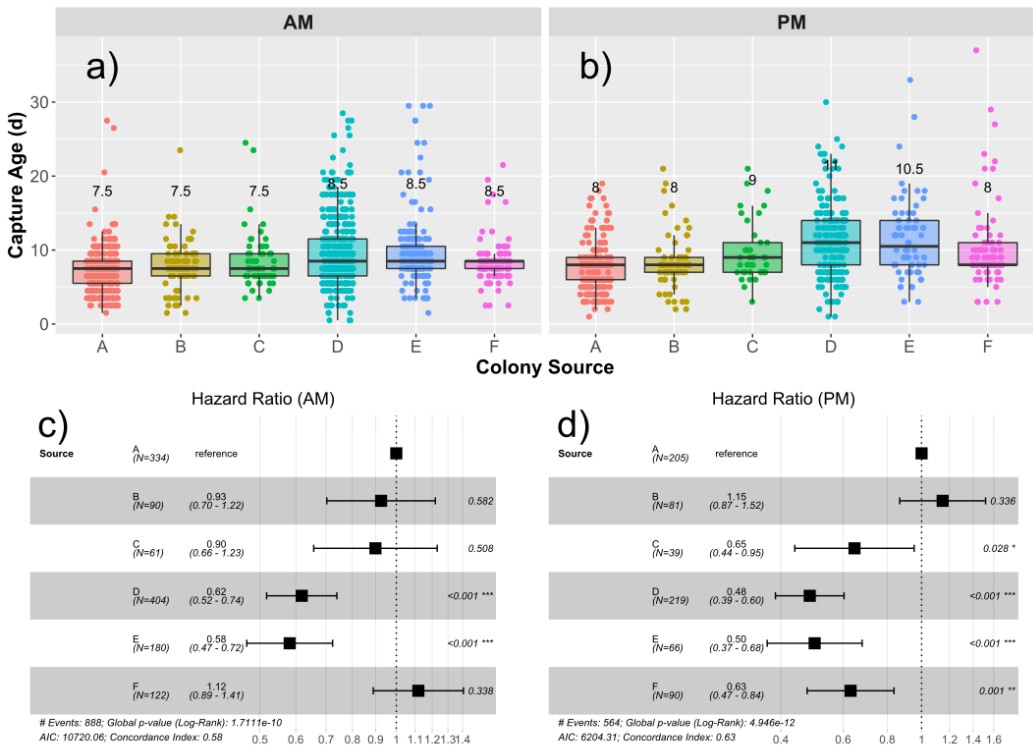

**Figure 2** **Drone capture ages by source and foster colonies.** Boxplots showing the distribution of capture ages in the AM (A) and PM (B) foster colony. Median age of capture ranged from 7.5–8.5 d in the AM colony and 8–11 d in the PM colony. Median capture age for each source is written over the box for each source colony. Cox proportional hazard ratios are presented in the forest plots for the AM (C) and PM (D) foster colonies separately. In both cases, Colony A was taken as the reference and the relative risks to be captured are presented for each subsequent source colony. Relative risks less than 1 represent a lower risk to be captured and correlate to a higher age of capture.

($t_{69} = 9.44$; $p < 0.0001$) body mass. Unexpectedly, drones gained $0.029 \pm 0.011$ (0.3%) mm ($t_{69} = -2.65$; $p = 0.001$) in head width, and gained $0.174 \pm 0.188$ (3.3%) mm in thorax width ($t_{69} = -9.27$; $p < 0.0001$) from emergence to capture. These differences were all significantly correlated with capture age such that drones captured at older ages lost more body mass (Spearman's Rho $= 0.35$; $p = 0.003$) and gained more head (Spearman's Rho $= -0.28$; $p = 0.020$) and thorax width (Spearman's Rho $= -0.26$; $p = 0.030$).

Finally, we tested whether emerged properties, source colony, and capture age was related to variation among the reproductive characteristics of drones, which are themselves subject to age-based ontogeny (*Metz & Tarpy, 2019*). We used a main-effects multiple regression model to test for effects on the variance in total sperm count (log-transformed), sperm viability (arcsine-transformed), and mucus gland length (mm). We found a whole-model significance for sperm count ($F_{8,60} = 5.05$; $p < 0.0001$; $r^2 = 0.32$) with a significant effect of capture age ($F_{1,60} = 11.19$; $p = 0.001$), source colony ($F_{4,60} = 5.37$; $p = 0.001$) and emerged body mass ($F_{1,60} = 5.27$; $p = 0.025$), but no significant effect of emerged head or thorax width (Fig. 4A). For sperm viability, the whole model was again significant

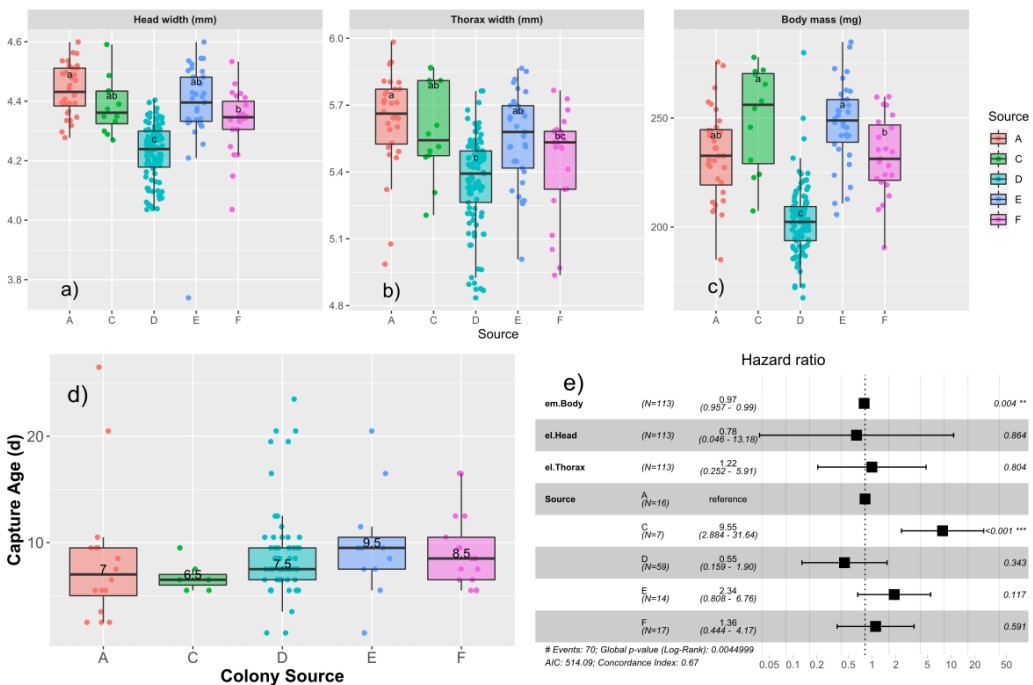

**Figure 3** **Relationships among age of capture, larval source colony, and emergent body mass.** Box-plots of the emergence parameters: head width (A), thorax width (B), and body mass (C) for each source colony. Tukey's significant groups ($p < 0.05$) are presented within each box. Note that drone emergence measures were only taken for drones installed into the AM foster colony. Median age of capture for each source (D) differs slightly in this subset of data relative to the ages of the expanded population presented in Fig. 2A and here ranged from 6.5–9.5 d. Cox proportional hazard ratios (E) show a significantly higher risk of flight for Colony C and a slight, but significant decrease in risk for each unit increase in body mass.

($F_{8,60} = 2.16$; $p = 0.01$; $r^2 = 0.17$) with capture age ($F_{1,60} = 9.26$; $p = 0.003$) and colony source ($F_{4,60} = 2.54$; $p = 0.049$) being the only significant factors (Fig. 4B). Finally for mucus gland length, the whole model was again significant ($F_{8,38} = 3.22$; $p = 0.007$; $r^2 = 0.28$) with capture age ($F_{1,38} = 8.02$; $p = 0.007$) and colony source ($F_{4,38} = 2.76$; $p = 0.042$) being the only significant factors.

# DISCUSSION

## Age of capture differs by colony source

The age at which drones attempted initial flight differed by both colony source and foster colony, with a positive interaction between the two. Drones from colony A were captured a median of 3–5 days earlier than the other colonies (Figs. 2A and 2B), suggesting that like for workers, genetics and rearing environment—which we cannot disambiguate in this study—both play a part in behavioral ontogeny. Because there was a source-by-foster colony interaction, we could not directly observe the effects of foster colony and therefore adult environment. However, the difference in median attempted flight age between the two foster colonies was on average one day, with drones from colony F being captured a median of 0.5 days earlier in the second foster colony, and drones from colony D being captured
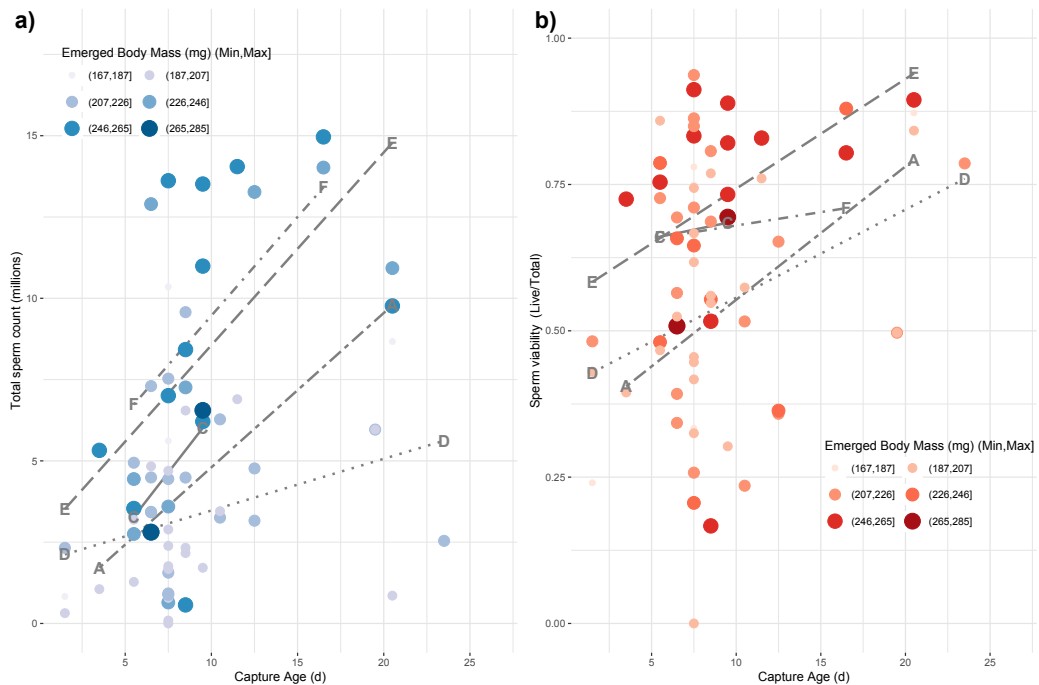

**Figure 4  Impact of age of capture and emerged body mass on captured sperm count.** The impacts of emerged mass and capture age are presented as causal factors for sperm count (A) and sperm viability (B) at capture based on main-effects, multiple regression model. While mass was modeled as a continuous variable, here it is binned into six categories for easier visualization with larger, more darkly colored points represent larger drones. Source colony was also significant in these models. Line types represent the relationship among capture age and sperm count for each colony, separately. Each colony is labeled at the minimum and maximum position for its respective line.

a median of 2.5 days later (Figs. 2A and 2B). This suggests that the 12-hour difference in introduction time between the two foster colonies is not sufficient to explain the variation in attempted flight time from one colony to the other. It is therefore likely that, similarly as in workers, both larval and adult colony environment elicits an effect on behavioral maturation (*Winston & Katz, 1982*). Our median likelihood for age of attempted flight was 8.5 days, about 2 days earlier than that reported in previous study (*Rueppell et al., 2005*). In addition to probable genetic and environmental variation based on evidence shown here, we also used a looser definition of "flight," since we included any drone that attempted to exit the hive entrance. Any drone capable of flight that was expelled from the nest could also likely be counted in the trap, as could drones attempting defecation and orientation flights, all of which would decrease our estimate. However, since the initiation of orientation flights, while occurring prior to full reproductive maturity, remains an indicator of being upon the precipice of conducting mating attempts, there is no reason to assume that small drones take more orientation flights than larger drones. It has been previously found that drones of varying size are subjected to differential treatment by nestmates (*Goins & Schneider, 2013*); such variation in treatment might serve as a mechanism for the variation in flight ontogeny.

### Drone emergence characteristics, age of capture, and fecundity

Drone emergence size (head width, thorax mass, and body mass) differed significantly among the source colonies. This was particularly true for colony C with colony F intermediate (Figs. 3A–3C). However, only drone body mass at emergence was significantly related to age of first attempted flight, such that drones that were less massive upon emergence were more likely to be captured at a younger age, even when accounting for source colony. Body size broadly trades off with the rate to achieve sexual maturity in many species (Morita & Fukuwaka, 2006; Stearns, 1992), and while we are not aware of a study specific to males, honey bee development to maturity differs among subspecies consistent with their overall adult size (reviewed in Nunes-Silva et al., 2006). Our results are consistent with the idea that the smaller individuals leave the nest earlier.

Smaller drones were also less fecund upon capture (Fig. 4). Drones often initiate flight prior to full reproductive maturity (Witherell, 1971). Our measurement paradigm—specifically measuring sperm parameters from the seminal vesicles rather than upon ejaculation—is intended in part to obviate the problems of sampling somewhat immature drones. Consequently, we do not know if the drones attempting flight at younger ages would be capable of normal mating behavior. Additionally, we do not know if the time from the first attempted flight to the time of full mating capability is different for drones of different size or is relatively developmentally constant. Although this would require detailed functional mating studies to determine, we surmise that because smaller drones are, if anything, developing faster than larger drones, we would not observe that smaller drones are attempting flight at a less reproductively mature state than their larger brethren.

### Causes and correlates of drone variation in emergence size and age of capture

In attempting to find an environmental cofactor, we measured colony-level mite loads and the mean cell sizes from which the drones emerged. Mite loads elicited a small but significant effect in the expected direction, with colonies that had higher mite counts producing less massive drones. This is similar to the effects of mite feeding shown in workers (e.g., Schneider & Drescher, 1987). Confusingly, however, cell size elicited an effect in the opposite direction, with smaller drones emerging from colonies with larger diameter cells. Despite the numerous examples of worker-cell produced drones in the literature being notably smaller (e.g., Gençer & Kahya, 2020), it appears that "natural" variation in drone cell size has little bearing on drone morphology. It is possible, though, that total cell volume rather than cell width is a better correlate with drone size, but we did not directly measure this. Moreover, while it is well known that the investment into brood production in general (although not specifically for drones) is a function of colony nutritional status (Ahmad et al., 2021; Brodschneider & Crailsheim, 2010; Hoover, Ovinge & Kearns, 2022; Noordyke & Ellis, 2021), we did not experimentally vary or quantify the food resources in the rearing or foster colonies. Manipulating the nutritional resources among colonies would be another variable worthy of future testing for effects on individual drone fecundity.

### Evolutionary rationale for variation in drone mating quality

The trait parameters of drones are consistent with the presence of pre- (*Koeniger et al., 2005*) and post-copulatory competition (*Baer, 2005*; *Liberti et al., 2019*), and indeed there appears to be a positive relationship between mating success and drone size (*Couvillon et al., 2010*). However, it is plausible that directional selection on individual drones may be counterbalanced by opposing selective pressures at the colony level (*Keller, 1999*). Specifically, there are likely energetic costs of rearing high-quality drones, inefficiencies in division of labor for nursing behavior, and colony-level regulation of larval rearing state and adult sustenance. Moreover, because the numerical sex ratio is so heavily male-biased, colonies may be adaptively favored to produce a wider range in drone sizes so that they can capitalize on rare mating opportunities. Combined with prior research, our results suggest that colonies make distinct resource commitments into their drone population and that these commitments have a long-term impact on the developmental ontogeny of their drones. At an individual level, drones of varying sizes may use different lifetime mating strategies (with larger drones making fewer mating flights with a higher likelihood of success but smaller drones attempting more flights each with a lower likelihood of success), although this intriguing possibility will require additional empirical investigation.

## ACKNOWLEDGEMENTS

The authors wish to thank the following lab members who participated in numerous late-night dissections and aided in the marking and collections of thousands of drones: Erin McDermott, Will Fowler, James Withrow, Jen Keller, Joe Milone, Nissa Coit, Dan Charbaneau, Lauren Russert, Claire Collins, Zacharias Everson, Ashley Rua, and Hannah Levenson.

### Funding

This work was funded by the U.S. Army Research Laboratory (No. W911NF1920306) and the USDA National Institute of Food and Agriculture (No. 2016-07962). The funders had no role in study design, data collection and analysis, decision to publish, or preparation of the manuscript.

### Grant Disclosures

The following grant information was disclosed by the authors:
US Army Research Laboratory:  W911NF1920306.
USDA National Institute of Food and Agriculture:  2016-07962.

### Competing Interests

The authors declare there are no competing interests.
## Author Contributions

- Bradley N. Metz conceived and designed the experiments, performed the experiments, analyzed the data, prepared figures and/or tables, authored or reviewed drafts of the article, and approved the final draft.
- David R. Tarpy conceived and designed the experiments, authored or reviewed drafts of the article, and approved the final draft.

## Data Availability

Raw data and code are available in the Supplementary Files.

## Supplemental Information

Supplemental information for this article can be found online at http://dx.doi.org/10.7717/peerj.13859#supplemental-information.

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
