# Peer review of "Variation in the reproductive quality of honey bee males affects their age of flight attempt"

_PeerJ, doi:10.7717/peerj.13859_

## Round 0.1 · original submission · Major Revisions

Both reviewers have suggested edits that require careful reanalyses of your data. I trust you will find that their suggestions will help improve the clarity of your study. If you do not agree, then please clearly justify your reasoning.

Reviewer 1 ·

Basic reporting

In general, the manuscript was easy to follow. It meets most of the requirements of the journal, but I had a few issues to be addressed:

I had some issues with a few complex sentences that were difficult to follow:
L85-87
L99-101
L101- Do you mean that you would expect a negative correlation between drone number and average drone size across colonies?

The Introduction was long and in some places contained unnecessary and distracting detail - the necessity of L59-66 was unclear, L66-79 seems like it could be more concise. I found myself wondering specifically about the previous study showing a reproductive advantage to large males - did they describe natural variation in size? Over what timeframe and colony sampling? Was there an idea of the range of size? There weren't many details on this study, but instead a lot of speculation about sources of variation in drone size that seem unlikely to be widespread in a natural context (queens laying in worker cells, eg). That seemed unnecessary. Similarly, parasitic feeding jumped into my mind at the outset of a source of body mass variation, and that played out here even though it received very light treatment throughout the manuscript. Perhaps you can streamline the introduction and paint a clearer picture about what is known about mass/size variation. Some of these other details could be moved to the discussion, where I would also like to see more about energetic trade-offs associated with adult body size variation and flight, larger animals have a higher mass-specific metabolic rate typically. Finally, I would like more discussion of investment in larval brood production as a function of nutritional stress. Under nutritional stress, are fewer drones produced? There is evidence that food limitation impacts worker bee physiology later in life (though maybe not body size). You could briefly mention these things and perhaps return in more depth later in the Discussion.

L75 - I don't fault the authors for this, as honey bee research has a deep history of problematic anthropomorphism, but I think the language "sloppy nursemaids" has a dual impact of sounding anthropomorphic and misogynist. Again, please don't take offense, I mean no judgement - we have a long way to go in improving language in honey bee research and behavior more broadly.

Experimental design

My biggest issue with your design and results approach is something that I think should be easy to fix - it has to do with how you report and assess "size". You report two measures of sclerotized body parts - head and thorax width. You report body mass, which encompasses both the absolute size of the animal and it's body condition (weight of soft parts, water, etc). Your result, that body mass is a predictor of emergence day while head/thorax width aren't, etc, suggest that it is in fact individual condition, not size per se, that is the best predictor of behavioral differences (and perhaps reproductive morphology differences). However, you can't definitively say that from how you report the data. I suggest calculating a true measure of condition, and re-analyzing the results following that measure (perhaps adding this to your current results if you'd like to). A condition measure is typically a measure of mass corrected for absolute body size. You would need first to determine the most accurate measure of absolute body size. There is usually a field standard, like intertegular distance, or leg length. Head width, which you already have, seems acceptable. One concern to resolve first, however, is why your measures of sclerotized body parts change over adult age, and moreover, change more the older the animal gets. I can't figure out why this would be. I recognize that animals on day 1 of adult emergence are not fully sclerotized, so their bodies could continue to grow. However, I know of no explanation for why body size would continue to change over the ~8-10 day timeframe of this study. How do you explain this? If you can resolve this problem, assessing body condition seems like the most relevant thing to do for this study. It is also a useful framework to think about trade-offs and reproductive strategies - the lighter males, at least in this study, were from colonies with more mites, which feed on developing bees (drones preferentially, as you know). The implications of variation in condition are different from those of variation in absolute body size.

Here you examine drone size variation over a very short time in the season. Throughout (including in the background) I don't see much context for the known variation in size and how that plays out over the season. I would guess there could be substantial variation with season due to many known factors (mite loads, nutritional stress) and unknown factors. Please provide more information about the role of season, and justify this short focal period.

Cell volume seems like it is a more critical determinant of size than just cell width. Is there any reason you focused on width?

Throughout the manuscript, your treatment of "foster colony" is very confusing. I think you had two colonies, one to which you introduced "AM" emerging drones, and one to which you introduced "PM" emerging drones. However, these colonies were sampled at the same time of day for drone collections. I strongly suggest changing how you label these foster colonies throughout. At present, you call them AM and PM, which is confusing. You could say "Foster Colony 1 with PM-emerging drones" or something. It is just hard to follow.

L234-236 - I think it is problematic to include the dead drones in the analysis of "flight date". These could have a variety of origins (including early death by natural causes) and it is extremely hard to interpret.

Minor things:
L154-155 - the way this is written, it sounds like you are saying you kept frames in the incubator for the duration of pre-adult development. Please clarify.
L167-168 - please explain why you did the mark recapturing with specific tags here. You do it in the Results, but it would help the reader to have it mentioned here too.
Did you account for weather variation from the time the drones were introduced, or was that an issue? That can have a big impact on even whether individuals emerge from the hive at all.
L225 - you may consider calling this "emergence" time and not "flight" time throughout because it is much more accurate as to what you actually measured. I expected some sort of measure of flight activity, duration, etc, based on the title and abstract.
L244 - you refer to the "reference colony", but it is unclear what this is and why you chose this colony. If this just means a statistical report of relative impacts of each source colony, please clarify. Any reason not to report post-hoc tests for, e.g., pairwise comparisons?

Validity of the findings

You should consider size and lifespan trade-offs in your discussion, especially since smaller males have a poorer condition. Moreover, there was very light discussion of Varroa mite feeding impacts and how those affect the interpretation of your results. Mites didn't co-evolve with apis mellifera, so that's interesting that they impact size and thus reproductive success. It is also worth considering how mite feeding may impact drone lifespan and flight capabilities. There are some studies in this area that aren't covered.

I'm not clear on your presentation of the results for mass and the relationship to mite count (L268). What is "234.54-1.06(mites)"? If this is an effect size estimate, can you please specify more clearly? (same comment applies to cell size)

L359 - I found this to be an overstatement. Even if queens mate 20x each, there is still a massive sex ratio bias towards males, and it is hard to argue this would not be a selective force, especially since colony reproductive potential is so different comparing the male and female contributions. The other components of this sentence confused me because they aren't related to drone competition. I think you are saying there are many forces that could influence drone size and constrain or weaken selection. That may be true, but don't overstate the possibility that all of these things sort of magically balance out!

·

Basic reporting

The manuscript by Metz et al is an interesting topic and the results add to the literature on this topic. The English is professional throughout. The manuscript could however be improved with addition of some relevant literature as outlined in the specific comments. The figures are of good quality however figure 4 could be improved. An additional figure covering sperm count and viability could improve the story.

Experimental design

The research question of this paper was: why do drones vary widely in their size and reproductive quality?
I feel the authors were not able to answer this question. Additionally the title: Variation in the size of honey bee males leads to different life history characteristics consistent with distinct mating strategy, is not covered in the results. While the authors set out to identify mating strategies, non such were discovered or described. Non of the results presented show evidence of differences in mating strategies. A different title may be more suitable.

I have some concerns regarding sample numbers and the large variation of drones sampled between different colonies. This needs to be at least addressed in the manuscript.

I have further concerns about the criteria of what is a drone attempting mating flights. A large number of drones was dead on day 0-1 and, in my opinion, should have been excluded from the results.

Some of the methods especially for correlation analysis could have been described in more detail. individual comments in the submission are highlighted.

Validity of the findings

Some of the data on sperm viability and count is not provided. Some statistics has not been described or reported. Conclusion is not substantiated by the results or literature.

Additional comments

General comments and questions to the authors.
The manuscript was an interesting read and I did enjoy the review, however I feel the paper could be improved. I do have some significant concerns, I would like the authors to address or respond to, these are outlined in the Pdf and above.
Additionally:
The number of tagged bees in Table 1 and sheet 1 in the supplement is different. Please consider explaining of correcting.
Generally cross-referencing the figures in the results would be helpful.

---

## Round 0.2 · Minor Revisions

I have some minor edit suggestions to improve clarity and writing:

Line 73, comma before ‘it may...’

Line 80: No need to have ‘more’ in italics. Same as line 128, 305, 306, and elsewhere in the ms

Line 105, patrol, rather than patroll

Line 111, Is the as before ‘compared to queens’ necessary?

Line 142: Spell out North Carolina, and add USA after it.

Line 149: Spell out RH to relative humidity.

Line 167, condition rather than conditions?

Line 172, comma after mating?

Line 201, remove additionally?

Line 244: Spell out HR at first use.

Figure 2, panels C and D. The alignment of p values on the right of the figures could be better.

Line 240/241: It is standard practice to provide stats anyways.

Line 252: Reorder to head, thorax, mass to align with figure where mass is panel C, not panel a?

Line 254, Cox capitialised as before at line 238?

Line 258, why report HR = 0.97, but as HR<1 at lines 244/245?

Line 324, commas around particularly for colony C?

Line 352: Notably smaller?

Various fonts and sizes appear to be used in the main text- please be consistent.

Line 422. Duff 2018 is a thesis and should be identified as such.

Please check that all scientific names are in italics in reference list. E.g., Winston 1982.

Reviewer 1 ·

Basic reporting

The Introduction is much improved and overall very clear and compelling. I had a few comments that I hope will be helpful for additional clarity. All other basic reporting requirements are met.

L47 – 48: add worker before population. Resource availability sort of, but season definitely, are environmental factors and not colony level factors. Perhaps reword the sentence to reflect that.

L 59 – it seems more accurate to say something like “their parental origin and how they are reared” (queen versus worker source of eggs isn’t really a rearing condition exactly)

L62 “laying” should be “lay”

L65 “and” should be “that”

L67 – “may be expressing” should be “may express”

L82 – 85 – this sentence was confusing to me and I’m not sure of the meaning. Perhaps reword?

L85 – 88 – I thought the way you explained this in the response to reviewer comments was much clearer than the wording here, which is confusing. For example, I think you are saying that drone fitness is impacted by selection on queen and worker traits through pleiotropy. Would this lead to “stochasticity” in drone development, or simply weaken the effects of selection on drones (weak selection in turn leads to greater variation in phenotype)? You go on to refer to the “queen-worker dichotomy”, which makes it seem like you are saying that something about the partitioning of maternal behaviors across these two castes has pleiotropic impacts on drones (an explanation that seems different and perhaps more complicated than the idea that selection on either caste impacts drones through pleiotropy). I think it would be good if you could come up with a way to word this sentence(s) a little more explicitly. One solution may be to put the sentence in L88-91 just before this.

L88 I don’t think you need the “-“ between “familial” and “social”

L92-93 – this topic sentence seems to confuse levels of analysis. Gene by environment interactions and genetic architecture problems like you described in the preceding paragraph are mechanisms that underlie adaptive responses, so they also have adaptive consequences. I think what you are arguing here is that variation in drone size could reflect a colony level reproductive strategy and that is that there may be balancing selection on drones as opposed to weakened directional selection (balancing selection could also maintain variation). Seems like the topic sentence could easily be clarified.

L104 – I appreciate the examples, but I think you could go straight into drone reproductive behavior and the possible avenues of alternative strategies (as you do about midway through the paragraph).

Experimental design

The authors addressed my previous concerns.

Validity of the findings

The authors addressed my previous concerns.

Additional comments

Very interesting study, and nicely written!

---

## Round 0.3 · accepted · Accept

Thank you for your thoughtful response to the minor edits/suggestions.